# A Prospective Investigation of Predictive Parameters for Preoperative Volume Assessment in Breast Reconstruction

**DOI:** 10.3390/jcm10225216

**Published:** 2021-11-09

**Authors:** Nicola Zingaretti, Giovanni Miotti, Carlo Alberto Maronese, Miriam Isola, Gianni Franco Guarneri, Roberta Albanese, Francesco De Francesco, Michele Riccio, Lorenzo Cereser, Chiara Zuiani, Pier Camillo Parodi

**Affiliations:** 1Clinic of Plastic and Reconstructive Surgery, Academic Hospital of Udine, Department of Medical Area (DAME), University of Udine, 33100 Udine, Italy; gio.miotti@gmail.com (G.M.); giannifranco.guarneri@asufc.sanita.fvg.it (G.F.G.); albaneseroberta16@gmail.com (R.A.); piercamillo.parodi@uniud.it (P.C.P.); 2Accademia del Lipofilling, Research and Training Center in Regenerative Surgery, 61025 Montelabbate, Italy; fran.defr@libero.it (F.D.F.); michele.riccio@ospedaliriuniti.marche.it (M.R.); 3UOC Dermatologia, Fondazione IRCCS Ca’ Granda Ospedale Maggiore Policlinico, 20162 Milano, Italy; carloalberto.maronese@outlook.it; 4Division of Medical Statistic, Department of Medicine, University of Udine, 33100 Udine, Italy; miriam.isola@uniud.it; 5Department of Reconstructive Surgery and hand Surgery, AOU “Ospedali Riuniti”, 60126 Ancona, Italy; 6Institute of Radiology, Department of Medicine, Academic Hospital of Udine, University of Udine, 33100 Udine, Italy; lorenzo.cereser@asufc.sanita.fvg.it (L.C.); chiara.zuiani@asufc.sanita.fvg.it (C.Z.)

**Keywords:** breast implant reconstruction, breast implant size, volume estimation, breast-V, conversion from weight to volume of mastectomy specimen, breast imaging, MRI

## Abstract

Preoperative breast volume estimation is very important for the success of the breast surgery. In this study four different breast volume determination methods were compared. The end-point of this prospective study was to evaluate the concordance between different modalities of breast volume assessment (MRI, BREAST-V, mastectomy specimen weight, conversion from weight to volume of mastectomy specimen) and the breast prosthetic volume implanted. The study enrolled 64 patients between 2017 and 2019, who had all been treated by the same surgeons for monolateral nipple–areola complex-sparing mastectomy and implant breast reconstruction. Only patients who had a breast reconstruction classified as “excellent” from an objective (BCCT.core software) and subjective (questionnaire) point of view at the 6-month interval after the operation were included in the study. Data analysis highlighted a strong correlation between the volumes of the chosen prostheses and the weights of mastectomy converted into volume, especially for patients with grades B and C parenchymal density. The values of the agreement between the volumes of the chosen prostheses and the assessments from MRI and BREAST -V proved to be lower than expected from the literature. None of the four studied methods presented any strong correlation with the initial breast width. Our results suggest that conversion from weight to volume of mastectomy specimen should be used to assist in determining the volume of the breast implant to be implanted. This method would help the reconstructive surgeon guide the choice of the most appropriate implant preoperatively.

## 1. Introduction

The goal of breast reconstruction is, in general, to achieve a shape and volume symmetry between the rebuilt breast and the contralateral one, taken as a model, despite knowing that the starting situation might often be marked by a light physiological asymmetry [1,2,3]. This leads, as a consequence, to the fact that the shape and volume of the prosthesis are crucial for the success of a breast reconstruction. Volume is actually the most critical variable: An error in the evaluation of the correct volume of the prosthesis may put at risk the final result [4]. From these observations arises the clinical question about the agreement on the volumetric evaluations in preoperative and intraoperative phases with the volume of the final prosthesis, the one that is used in the end.

Having a reliable and ready-to-use instrument to evaluate the breast volume would make it possible to optimize the planning of the surgical activity and would lead to better standardization of the decision-making process related to the prosthesis volume with a consequent saving of time and resources during the surgical operation.

Some studies provide evidence that intraoperative measurement of the weight of mastectomy specimens accurately reflects the breast volume, and this simple approach can be used to guide implant selection during breast reconstruction [5,6,7].

Recognizing the need for a fast and easy-to-use assessment device, based on easy-to-get data, Longo et al. developed a formula for volume assessment in small to large breasts based on anthropomorphic values called BREAST-V [8]. This study researched the correlation between the mastectomy specimen weight (MW) and some easy-to-measure parameters considered to be the best predictors for breast volume. BREAST-V is a fast and easy-to-use device, based on easy-to-get data, and is also available as an app for smartphones and tablets. The authors mention that this model shows an estimated expected absolute error of about 90 g and a relative error of about 18%, and, although it is suitable for small, medium, and large breasts, has a disadvantage in that it can be used only for ptotic or pseudoptotic breasts, which gives a positive value for inframammary fold-to-fold projection distance. 

In our clinical practice we have found that BREAST-V predictions have not always proven to be reliable; in particular, we have noticed the sensitivity of the preoperative assessment is reduced when breasts are quite large. 

Nowadays, conversion from weight to volume mastectomy (MWCV) has been considered one of the most cited technique for calculating the ideal implant volume for breast reconstruction. Lee et al. investigated the relationship between breast tissue weight and volume by assigning subjects into different groups according to breast density by using BI-RADS. They proposed a simpler and accurate method for measuring breast tissue volume by studying the relationship between the weight and volume of the excised tissue according to density [9]. 

The aim of our work was to evaluate the agreement between some methods for calculating the volumes of the implanted breasts (MRI, BREAST-V, MW, MWCV).

## 2. Materials and Methods

A prospective observational study was conducted on patients with primary breast cancer who underwent monolateral nipple–areola complex-sparing mastectomy (NSM) and breast implant reconstruction from January 2017 to June 2019. 

Inclusion criteria included patients whose aesthetic result was considered "excellent" from both objective (evaluation through BCCT.core software, version 3.1, Inesc Tec, Porto, Portugal) [10] and subjective (overall score equal to or greater than 3.5 in a questionnaire filled out by the patient) points of view 6 months after the reconstructive surgery.

Exclusion criteria included patients with postoperative complications requiring a further operation, undergoing postoperative radiotherapy, in need of contralateral symmetrization, previously having undergone other breast surgery, undergoing reconstruction with prepectoral implant placement, or another reconstructive surgical strategy that may not be the implant of the final prosthesis.

### 2.1. Preoperative Evaluation

The three anthropometric data required by the BREAST-V formula were measured in all the patients. Distances were taken as reported in the original article by the investigator himself by using a common tape measure [8]. The volumetric assessment was calculated using such measures through the BREAST-V application. 

Measurement of the parameter of the breast width (from the most lateral to the most medial point of the breast along a line passing through the nipple) was also taken. Thanks to the data obtained with the breast MRI, the volume of the rebuilt breast was assessed according to the methods outlined by Itsukuge et al. [11]. The evaluation was made by the same radiologist using the software TeraRecon iNtuition (AQI Viewer, Terarecon Inc, Foster City, CA, USA).

### 2.2. Operating Procedure and Intraoperative Assessments

On the day of surgery, each patient, in supine position and with general anesthesia intubation, underwent a NSM. After the mastectomy phase was over, the reconstructive plastic surgeon placed the final prosthesis and determined the correct volume by using adequate sizers and intraoperative assessment of the patient in a sitting position (45 degrees).

In all patients, the implants were anatomic and textured (CPG^TM^ Gel Breast Implant Cohesive III, Mentor Medical Systems, Irvine, CA, USA) and were used to perform a partial retropectoral reconstruction, in which the superior part of the breast implant is covered with the pectoralis major muscle, while the inferior part is covered with an acellular dermal matrix sling (Veritas, Synovis Surgical Innovations, St. Paul, MN, USA).

The implant was chosen by the surgeon independently; its choice was not influenced either by the BREAST-V application, MWCV, or MRI. 

At the end of the mastectomy, the operated mass was weighed by a digital scale (KERN 440–51N).

Subsequently, taking into account the patient’s BIRADS classification and using the conversion formulas suggested by Lee et al. [9], the weight of the mastectomy was converted into volume (MWCV).

The volumetric evaluations referring to the operated breasts of the enrolled patients and obtained during the preoperative and perioperative phase through BREAST-V application, MRI, and the MWCV were compared with the volume of the implanted prosthesis (considered to be the correct referring value in order to obtain an “excellent” reconstruction). Then calculation of the agreement (among each of the above-mentioned three volumetric assessments and the volume of the final implant) was made along with the correlation (between the width of the breast and the volumes’ differences assessed by each of the aforementioned three methods) for assessing the volume when compared with the volume of the final implants.

### 2.3. Post-Operative Evaluations

After 6 months from the operation, which is the necessary period of time for a first stabilization of the outcome, the patients were photographed and evaluated objectively during the routine post-operative check-ups. The objective assessment was performed using the BCCT.core software (3.1 version; Inesc Tec, Porto, Portugal) [10,12] and using clinical digital pictures (Nikon D750 + af-s nikkor 24–70 mm f/2.8G ED) taken frontally with the patient in an orthostatic position and with adducted arms.

BCCT.core software, considering symmetry chromatic alterations and scars, provides a complete aesthetic evaluation summarized in four possible results: poor, fair, good, or excellent [10,12] (Figure 1A,B).

Patients were asked to fill in a questionnaire in the follow-up period (6 months after surgery) to evaluate the outcomes of reconstructive surgery. For each parameter (five), patients gave their evaluation using a 1–4 scale, with 4 being excellent, 3 good, 2 sufficient, and 1 insufficient [1].

### 2.4. Statistical Analysis

Qualitative variables were summarized with media and standard deviation or median and range, as far as their distribution was concerned. For qualitative variables, the frequency distribution was calculated. Lin’s Concordance Correlation Coefficient (CCC) method was used to compare two measurements of the same variable [13]. This method measures the concordance of continuous variables measured by two operators or with two methods or at different times. 

Lin’s concordance correlation coefficient was evaluated as follows: values from 0 to 0.20 as poor agreement, from 0.21 to 0.40 as sufficient agreement, from 0.41 to 0.60 as moderate agreement, from 0.61 to 0.80 as substantial agreement, and from 0.81 to 1 as almost excellent agreement.

Limits of agreement were described by Bland and Altman analysis.

A Pearson or Spearman coefficient correlation was used to assess the correlation between MW implant volume and between the breast width and the difference of the volumes assessed by all the three methods compared with the one of the final implant actually used in the unilateral breast reconstructions after NSM.

Finally, we analyzed the composition of the obtained volumes with the three methods, combining two subgroups according to the volume of the implanted prosthesis (less than 300 cc or greater than 300 cc) and combining two subgroups according to parenchymal density (class BIRADS A and class BIRADS D parenchymal density or class BIRADS B and class BIRADS C parenchymal density). The threshold for statistical significance was set at *p*-values < 0.05.

## 3. Results

Patients’ demographics are reported in Table 1. 

The correlation between the volumetric assessments obtained from the mastectomy weight without conversion and the implant volume was 0.90 (95% CI: 0.84–0.95). This value was considered as a good correlation.

Lin’s CCC between the volumetric assessments obtained by MRI and the implant volume was 0.87 (95% CI: 0.81–0.93). This value was considered as an almost perfect agreement (Figure 2a).

Lin’s CCC between the volumetric assessments obtained with BREAST-V and the volume of the final implant actually chosen was 0.65 (95% CI: 0.53–0.76). This value was considered as substantial agreement (Figure 2b).

Finally, the agreement between the volumetric assessments obtained from the MWCV and the volume of the implant used was 0.89 (95% CI: 0.83–0.94); it was considered an almost perfect agreement (Figure 2c).

Spearman’s correlation coefficient *r* (with Bonferroni correction) between the breast width and the difference of the volumes assessed by all the three methods compared with the one of the final implant turned out to be quite weak and, in any case, was without statistical significance. 

The agreement of the volume obtained with the three methods compared with implant volume in the subgroup of the patients with volume below 300 cc (“low implant volume”) was 0.39 (95% CI: 0.15–0.64) for BREAST-V assessments, 0.62 (95% CI: 0.44–0.80) for MRI assessments, and finally 0.81 (95% CI: 0.70–0.91) for MWCV assessments. The agreement of the volumes obtained with the three methods compared with implant volume in the subgroup of the patients with volume above 300 cc (“high implant volume”) was 0.16 (95% CI: −0.03–0.36) for the BREAST-V assessments, 0.61 (95% CI: 0.36–0.86) for the MRI assessments, and, finally, 0.62 (95% CI: 0.37–0.88) for MWCV assessments.

The agreement between the volumes obtained with MRI methods and implant volume in the subgroup of the parenchymal density classes BIRADS B and C was 0.82 (95% CI: 0.74–0.91), whereas for the subgroup of parenchymal density of classes BIRADS A and D it was 0.91 (95% CI: 0.83–1.00).

The concordance between the prosthesis volume and the MWCV, when parenchymal density was classes BIRADS B and C, was 0.93 (95% CI: 0.89–0.97), but 0.84 (95% CI: 0.69–0.99) for the subgroup with classes BIRADS A and D parenchymal density.

Patients’ self-assessment is described in Table 2: Higher scores were achieved for questions 1 and 2. The overall average score (question 5) was 3.68 (SD: 0.14).

## 4. Discussion

Although breast implant volume estimation is important, nowadays there is still no commonly accepted standard method in order to determine the best breast implant size to use. Most surgeons are used to employing sizers, which are only indicated for single use for temporary insertion intraoperatively to evaluate the volume of the breast implant to be implanted. However, the use of one or more sizers shows some disadvantages, such as longer operating times and higher risks of infections and of capsular contracture [14,15]. In addition, some breast reconstruction centers may only have a limited number of sizers to use; therefore, this might invalidate the choice of the final implant volume.

Publications in scientific literature describe different methods to achieve direct assessment of the breast volume and many studies tried to provide a more evidence-based approach in order to help selecting the prosthetic implant and to use this information to better plan the preoperative phase of the reconstruction [7,8,9,10,15,16,17,18,19,20,21,22]. 

However, an accurate evaluation of such a parameter is difficult to make because the breast has a complex, three-dimensional shape, which may vary according to the patient’s position and because it may be complicated to identify its border with the thoracic wall in a precise and reproducible way.

This pilot study took into consideration the four breast volume assessment methods that are more widely present in scientific literature: a diagnostic imaging investigation (MRI), which is referred to as the most accurate one [16], an anthropometric model (BREAST-V) [8], the mastectomy weight without conversion (MW), and, finally, the MWCV in consideration of the breast parenchymal density, a measuring method considered by other authors to be the gold standard [9]. 

The methodological novelty of our study lies in setting as reference value neither the mastectomy weight nor its volume, but the volume of the implanted prosthesis. By doing this, even if it is not an interventional study, the resulting assessments were evaluated for whether or not they could lead to a correct reconstruction.

The BCCT.core software, an important toll for the assessment of aesthetic outcomes post unilateral breast reconstructive surgery [20], was used to enroll only patients in which the implant used gave them an "excellent" reconstruction.

Data analysis highlighted a strong correlation between the volumes of the selected prostheses and the MWCV. Contrary to expectations, the MWCV turned out to be the best predicting factor for implant volume (CCC = 0.89). 

Estimations obtained with MRI turned out to be lower (CCC = 0.84) than the literature would have led us to expect [15,23]; however, this outcome was most likely due to the need for more training in executing the evaluations and for more standardization of the procedure [9]. The assessments from MRI turned out to be, on average, higher than the expected values. 

The agreement between the evaluations by BREAST-V and the selected prosthetic implants, despite proving to be substantial (CCC = 0.65), turned out to be less than the CCC obtained with the mastectomy weight converted into volume, and this has raised some doubts about this quick preoperative strategy. The assessments from BREAST-V turned out to be, on average, lower than the expected values. 

The impression, resulting from clinical observation, that BREAST-V method is less accurate when assessing particularly large breasts was confirmed upon discovering the presence of a correlation (Spearman’s *r* = 0.01) between the width values of the examined breasts and the errors that occurred in assessing the volume of the final prostheses. However, none of the three studied methods could meet the criteria suggested by Losken and Probst in terms of the minimum characteristics a measuring system for the breast volume should have to be deemed adequate for the needs of the modern clinical practice [21,22]. 

Among the cases that were objects of this study, the most numerous concerned small breasts (mastectomy weight less than 300 g in 69 % of the cases). The reason behind this is linked to the fact that it is, in general, preferred to address women with breasts of small to medium size to a direct-to-implant strategy without contralateral symmetrization. This may have had an influence on some of the results obtained.

This study encourages the use of the MWCV to guide the surgeons in the choice of the prosthetic volume. When the parenchymal density was equal to B or C, the MWCV was very close to the volume of the final implant. On the other hand, when the parenchymal density was equal to A or D, the MWCV deviated more from the original value. MRI evaluations, despite proving to be a solid method, obtained poorer results than those obtained with the assessments made with the conversion of the weight into volume. Finally, the role of the BREAST-V application needs further clarification, as it turned out not to be as reliably predictable as one could have expected considering the premises.

As expected, none of the three methods used for the volumetric assessment (BREAST-V application, MRI assessments, MWCV) showed any concordance with breast width. This was only to be expected because the aim of this study was to analyze the prosthesis volume, not its shape. Implants with equal volume might have different shapes and, therefore, different widths and projections.

In this sense, it is necessary to perform further studies, allowing variation of the formulas currently used for volumetric calculation and taking into account the width of the starting breast.

It is interesting, though, that the outcome was obtained by dividing the population into subgroups according to the volume of the prosthesis used (above or below 300 cc) and according to the breast parenchymal density (classes BIRADS A and D compared with classes BIRADS B and C). 

The BREAST-V app seems to be more reliable when the implant volume is less than 300 cc; the MWCV turns out to be the most reliable method for both subgroups, even if its reliability tends to decrease as the volume of the prosthesis used increases. 

Furthermore, the method of the MWCV seems to be the best option when the breast has a class BIRADS B or C parenchymal density, but, for class BIRADS A or D parenchymal density, the best method is the MRI, even though it should be stated that further studies with a larger sample are required in order to confirm the results so far obtained.

### Limitations of the study

This study has some limitations that should be pointed out:

(1) The physiological asymmetry of the two breasts. The reconstruction of the operated breast, taking as a model the contralateral one, always bearing in mind the condition of the physiological asymmetry between the two, and the subsequent use of the volume of the selected implant as a reference bring their own biases into the assessment of the accuracy of BREAST-V method and of the one obtained with MRI, since they both are preoperative evaluations made on the breast to be operated.

(2) Although mastectomy was made by the same senologist surgeon and breast reconstruction by the same plastic surgeon, the reproducibility of the entity of the resection of the adipo-glandular tissue in a NSM (and so the thickness of the mastectomy flap) might vary from patient to patient.

(3) Two-dimensional analysis of clinical photographs may have errors in measurement.

## 5. Conclusions

In this paper, we presented our experience in the choice of the preoperative and intraoperative prosthetic volume. The BREAST-V application turned out to be a reliable method to assess the volume for small breasts. 

For breasts with medium parenchymal density (class BIRADS B or C), the best method to assess breast volume of the implant to be used was the conversion of the weight into volume. MRI evaluations seemed to be a reliable method for breasts with class BIRADS A or D parenchymal density. 

We do believe that further studies with larger series, longer follow-up, and control groups are necessary to definitively confirm our results in the long term.

## Figures and Tables

**Figure 1 jcm-10-05216-f001:**
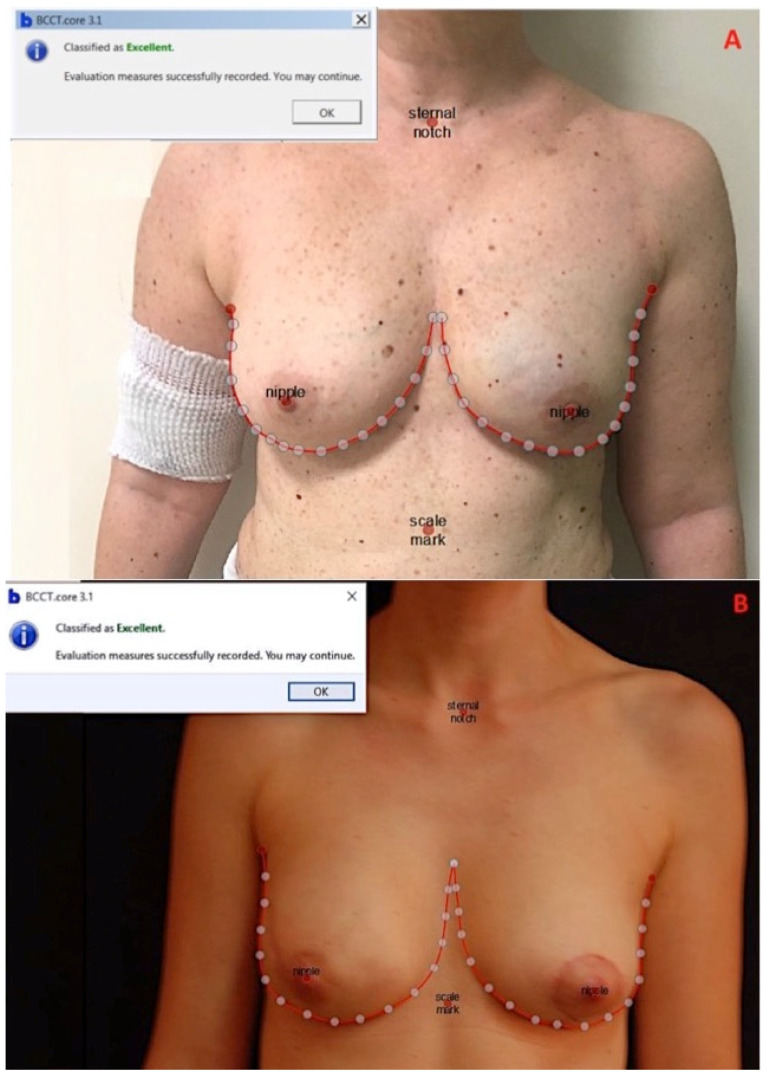
(**A**) Patient after 7 months from NAC-sparing left mastectomy and immediate reconstruction with 290 cc Mentor CPG 312 submuscular prothesis and ADM (dual-plane). This reconstruction was rated as excellent by BCCT.core software and 3.75 (the overall average score) by the questionnaire. (**B**) Patient after 10 months from NAC-sparing right mastectomy and immediate reconstruction with 170 cc Mentor CPG 312 submuscular prothesis and ADM (dual-plane). This reconstruction was rated as excellent by BCCT.core and 3.5 (the overall average score) by the questionnaire.

**Figure 2 jcm-10-05216-f002:**
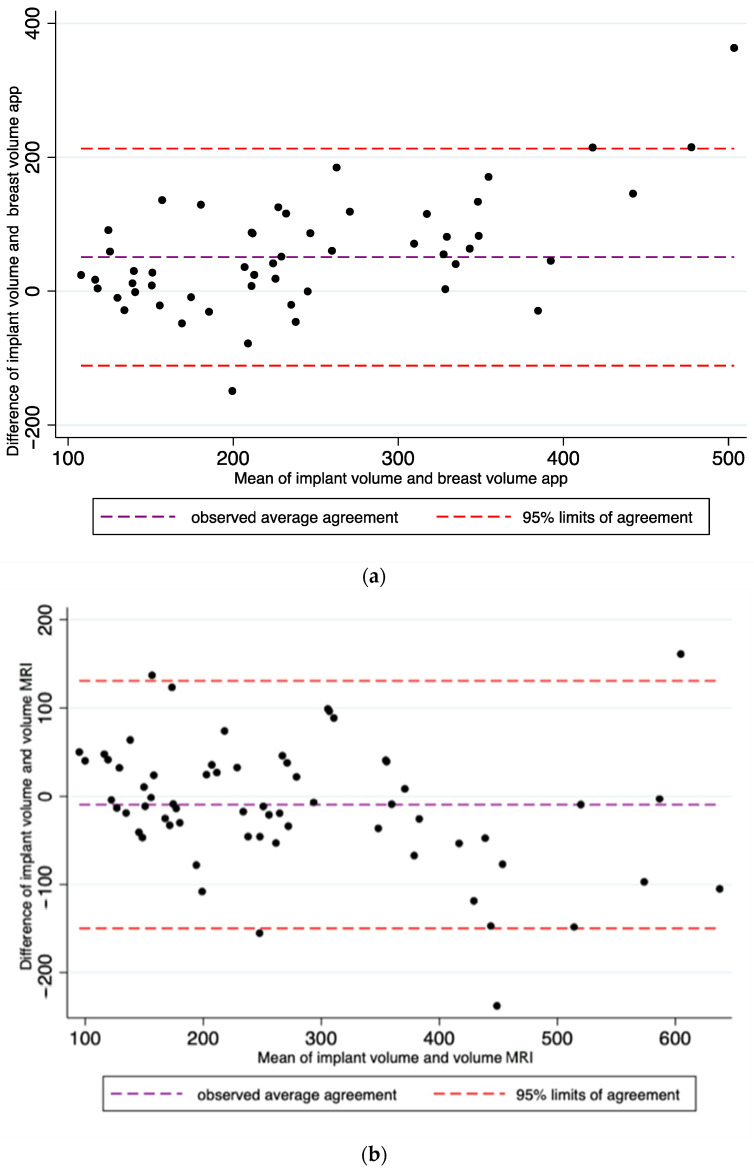
In these figures, the Bland–Altman analysis illustrates the limits of agreement between the volumes of the selected implants and BREAST-V app (**a**), between the volumes of the selected implants and MRI (**b**), and between the volumes of the selected implants and mastectomy weight converted into volumes (**c**).

**Table 1 jcm-10-05216-t001:** Patient demographics.

**Number patients**	64
**Age** (years)	
Median (range)	49 (35–75)
**Side**	Number (%)
Left	40 (62.50)
Right	24 (37.50)
**Sternal notch—nipple distance** (cm)	
Mean (SD)	20.84 (1.90)
**Inframammary fold—nipple distance** (cm)	
Median (range)	7.5 (4–12)
**Inframammary fold–fold projection distance** (cm)	
Mean (SD)	4.93 (1.98)
**Breast width** (cm)	
Mean (SD)	12.33 (1.66)
**Breast-V app** (cm^3^)	
Mean (SD)	218.83 (84.02)
**Volume MRI** (cm^3^)	
Median (range)	254.50 (69.80–690)
**Mastectomy weight** (g)	
Median (range)	213.55 (83.30–815.70)
**Parenchymal density**	Number (%)
A	6 (9.38)
B	24 (37.50)
C	24 (37.50)
D	10 (15.62)
**Conversion from weight to volume** (cm^3^)	
Median (range)	228.70 (88.40–696.30)
**Conversion from weight to volume** (cm^3^)	
Median (range)	228.70 (88.40–696.30)
**Implant Volume** (cm^3^)	
Median (range)	240 (120–685)
**Implant Type**	Number (%)
**CPG 311**	6 (9.38)
**CPG 312**	31 (48.44)
**CPG 321**	16 ( 25.00)
**CPG 322**	7 (10.94)
**CPG 323**	4 ( 6.25)

SD: standard deviation, Breast-V app: Breast-V application, MRI: magnetic resonance imaging, CPG: Contour Profile Gel.

**Table 2 jcm-10-05216-t002:** Patient self-assessment evaluation 6 months after breast reconstruction.

Questions	Post-Operative (6 Months After) Mean Score (SD)
1. Shape of breast with bra	3.88 (0.22)
2. Breast symmetry with bra	3.76 (0.11)
3. Shape of breast without bra	3.67 (0.16)
4. Breast symmetry without bra	3.51 (0.19)
5. Overall average score	3.68 (0.14)

A 4-grade scale was used in each category (4, excellent; 3, good; 2, sufficient; 1, insufficient).

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
