# Peer review of "A Prospective Investigation of Predictive Parameters for Preoperative Volume Assessment in Breast Reconstruction"

_jcm, 2021, doi:10.3390/jcm10225216_

Round 1

Reviewer 1 Report

This study aimed at investigating the association between methods for calculating the volumes of the implanted breasts.

Please clarify the laterality of breast cancer and reconstruction. Were they all bilateral? Prophylactic or therapeutic?

Please provide more details about BREAST-V formula (using which parameters), BCCT.core and MWCV. Are they widely used in the clinical settings?

Please add short title of Figure 1.

Please add more limitations inherent with this study. 2 dimensional analysis of clinical photographs may have errors in measurement.

Author Response

Reviewers' comments:
Reviewer 1
This study aimed at investigating the association between methods for calculating the volumes of the implanted breasts.

Please clarify the laterality of breast cancer and reconstruction. Were they all bilateral? Prophylactic or therapeutic?

All patients (64) with primary monolateral breast cancer underwent to therapeutic monolateral nipple sparing mastectomy (lines 78-80). As reported in table 1, 40 patients underwent left nipple sparing mastectomy and 24 right nipple sparing mastectomy and immediate implant reconstruction.

Please provide more details about BREAST-V formula (using which parameters), BCCT.core and MWCV. Are they widely used in the clinical settings?

This pilot study takes into consideration the four breast volume assessment methods that are more widely present in scientific literature (lines 230-233,238-243).

The authors of BREAST-V (Longo B et al. The breast-V: a uniying predictive formula for volume assessment in small, medium, and large breasts. Plastic Reconstr Surg. 2013; 132: 1e-7e) aimed to develop a unifying predictive formula for volume assessment in small to large breasts based on anthropomorphic values (lines 55-60, 91-94): BREAST−V = −231.66 + 0.5747 × (sternal notch to nipple distance)2 + 18.5478 × (fold to fold projection point distance ) + 14.5087 × (fold to nipple distance).

The authors released an app entitled BREAST-V for both iOS devices and Android smartphones available to be downloaded for free on Apple’s online App Store and Google Play Store (lines 58-60). The developers reported (during the Italian Society of Plastic Surgery Congress in 2016) a great number of plastic surgeons downloaded the app.

We have used BREAST-V formula in our clinical practice (lines 65-67). Notwithstanding its accuracy, it has to be kept in mind that the BREAST-V must not be considered as an alternative but as a complementary device to be combined with the experience, aptitude, and ability of the surgeon approaching the decisional process of every breast-shaping procedure.

The BCCT.core software was developed in 2007. It is a user friendly semi-automatic objective tool that evaluates asymmetry, color differences and scar visibility using patient’s digital pictures (lines 135-137).

During the last 14 years the BCCT.core software has made proof of its value through requests and publications. As reported by Preuss J et al (Preuss J, Lester L, Saunders C. BCCT.core – Can a computer program be used for the assessment of aesthetic outcome after breast reconstructive surgery? Breast 2012; 21:597-600), BCCT.core demonstrated correlation with the Harris Scale for measuring aesthetic outcome for unilateral surgery and suggests potential use for BCCT.core to measure aesthetics for breast reconstruction after mastectomy (lines 248-250).

We used this software as a tool to choose patients to enroll (lines 81-82).

Conversion from weight to volume of mastectomy specimen (MWCV) aimed to devise a simpler and more accurate method for measuring breast tissue volume by studying the relationship between the weight and volume of the excised tissue according to density (lines 68-70).

Lee and colleagues designed equations to calculate the breast volume on the basis of the breast weight, incorporating adjustment for breast density (lines 70-74).

The regression equations are as follows:

group BI-RADS A: V = 1.218 × W+7.45 (V: volume, W: weight).

group BI-RADS B: V = 1.036 × W + 10.36.

group BI-RADS C: V = 0.969 × W−7.47.

group BI-RADS D: V = 0.871 × W−14.13.

Nowadays it is one of the four breast volume assessment methods that are more widely present in scientific literature (lines 238-243).

Please add short title of Figure 1.

We edited the title of Figure 1.

Please add more limitations inherent with this study. 2 dimensional analysis of clinical photographs may have errors in measurement.

We agree with the reviewer. The main criticism to the BCCT.core software was and still is related to the fact that it does not include oblique or lateral views.

We used BCCT.core software because it is a valid and efficient method of evaluation of cosmetic results and is validated in various works in literature (references 11, 12 and 24).

As reported by Cardoso MJ et al (Turning subjective into objective: the BCCT.core software for evaluation of cosmetic results in breast cancer conservative treatment. Breast; 2007; 16:456-461),

it has the potential to become a gold standard method for assessment of breast cosmesis in clinical trials, as it can be used simultaneously by a panel of observers from different parts of the world to provide more reliable assessments than has been possible previously. For this reason, this software has allowed us to make an objective evaluation of the breast symmetry and in this way to find a reference value (volume of the chosen prosthesis) to be used to compare the various parameters taken into account in this article.

We agree that volume information is important and can add another dimension to the evaluation but 3D evaluation continues to be very difficult to generalize due to acquisition conditions and cost (Eder M, Waldenfels VF, Swobodnik A et al. Objective breast symmetry evaluation using 3-D surface imaging. Breast; 2021; 21: 152-158).

This information has been added in the “limitation of the study” section (line 316)

Reviewer 2 Report

Some English language needs to be revised specially in the introduction 

Author Response

Reviewer 2:

Some English language needs to be revised specially in the introduction

English language has been polished both in the abstract & in the text sections. Major corrections are written in red.

Reviewer 3 Report

The work deals with the possible predictability of the ideal implant size in implant reconstruction. The methodological approach is interesting, and the question is fundamentally attractive. However, the paper has some linguistic and grammatical deficiencies and should be revised by a professional language corrector. Some formulations are somewhat unfortunate or unclear. In terms of content, the techniques of volume estimation should be better explained. The correlation of the MW with implant size is unfortunately not discussed any further, so that it remains unclear what added value the MWCV actually has. The conversion factors are unfortunately also missing. Dual plane is a term from primary augmentation. Demolition phase is an inappropriate term. There are some minor errors, as BIRADS C and class BIRADS C in line 169. I would recommend resubmitting the manuscript after revising the language and content.

Author Response

Reviewer 3:

The work deals with the possible predictability of the ideal implant size in implant reconstruction. The methodological approach is interesting, and the question is fundamentally attractive. However, the paper has some linguistic and grammatical deficiencies and should be revised by a professional language corrector.

The text has been shortened and repetitions avoided. English language has been polished both in the abstract & in the text sections. The English language and the spell check was reviewed by a mother tongue speaker. Major corrections are written in red.

Some formulations are somewhat unfortunate or unclear. In terms of content, the techniques of volume estimation should be better explained. The correlation of the MW with implant size is unfortunately not discussed any further, so that it remains unclear what added value the MWCV actually has. The conversion factors are unfortunately also missing. Dual plane is a term from primary augmentation. Demolition phase is an inappropriate term. There are some minor errors, as BIRADS C and class BIRADS C in line 169. I would recommend resubmitting the manuscript after revising the language and content.

We agree with the reviewer.

We edited the text as adviced by the reviewer (lines 51-53, 68-74, 103, 109, 171)

Round 2

Reviewer 1 Report

I appreciate the author's response.